# Blood Levels of Neuropeptide 26RFa in Relation to Anxiety and Aggressive Behavior in Humans—An Exploratory Study

**DOI:** 10.3390/brainsci13020237

**Published:** 2023-01-31

**Authors:** Henning Værøy, Saloua Takhlidjt, Yamina Cherifi, Emilie Lahaye, Nicolas Chartrel, Serguei O. Fetissov

**Affiliations:** 1Department of Psychiatric Research, Akershus University Hospital, N-1478 Nordbyhagen, Norway; 2Regulatory Peptides-Energy Metabolism and Motivated Behavior Team, Neuroendocrine, Endocrine and Germinal Differentiation and Communication Laboratory, Inserm UMR1239, University of Rouen Normandie, 76000 Rouen, France

**Keywords:** RFamide, QRFP, motivated behavior, emotion, mood, hostility

## Abstract

26RFa, also referred to as QRFP, is a hypothalamic neuropeptide mainly known for its role in the regulation of appetite and glucose metabolism. Its possible relevance to emotional regulation is largely unexplored. To address this, in the present exploratory study, we analyzed the plasma concentrations of 26RFa in humans characterized by different levels of anxiety and aggressive behavior. For this purpose, the study included 13 prison inmates who have committed violent crimes and 19 age-matched healthy men from the general population as controls. Anxiety, depression and aggressive behavior were evaluated in both groups using standard questionnaires. The inmate group was characterized by increased aggression and anxiety compared to the controls. We found that the mean plasma levels of 26RFa did not significantly differ between the inmates and the controls. However, several high outliers were present only in the inmate group. The plasma levels of 26RFa correlated positively with the anxiety scores in all the studied subjects and controls. After removing the high outliers in the inmate group, positive correlations of 26RFa with anxiety and a subscale of hostility in the aggression scale were also recorded in this group. No significant correlations of 26RFa with depression scores or other parameters of aggressive behavior were found. Thus, the present results did not support an involvement of 26RFa in aggressive behavior in humans but pointed to a link between this neuropeptide and anxiety. Nevertheless, considering the exploratory nature of the present study, this conclusion should be verified in a larger cohort, including the clinical degree of anxiety.

## 1. Introduction

26RFa is a 26-amino acid peptide discovered in 2003 by three groups, including our team [1,2,3]. 26RFa and its N-terminus elongated 43-amino acid form, called QRFP or pyroglutamylated RFamide peptide, are the cognate ligands of the human orphan receptor GPR103 [1,3]. Additionally, 26RFa is a member of the RFamide-related peptide (RFRP) family characterized by the same Arg-Phe-NH_2_ amino acid signature at their C-terminus and is involved in the modulation of various neuronal and neuroendocrine functions [4,5,6]. In the brain, 26RFa’s precursor mRNA is highly expressed in the regions controlling energy metabolism and appetite, such as in the ventromedial hypothalamus (VMH) and the lateral hypothalamic area (LHA) [7]. Indeed, several earlier studies showed that 26RFa has a central orexigenic effect [8]. Moreover, 26RFa plays a key role in mediating the hypoglycemic effect of insulin in the brain [9]. Hypothalamic expression of 26RFa, in particular in the VMH region, which harbors the ‘aggression center’ in the ventrolateral part of the ventromedial nucleus (VMN) [10], suggests that it may also be relevant to aggressive behavior. Furthermore, the VMN displays a high level of GPR103 mRNA [11]. A modulatory effect of RFRP, another peptide of the RFamide family, on activation of the VMN neurons and reduction of aggressive behavior was indeed shown in mice [12]. However, in contrast to 26RFa, RFRP was shown to decrease the food intake in mice [13]. The possible relevance of 26RFa to the regulation of mood and emotion is also largely unknown and is presently limited to reports showing a reduction of anxiety-like behavior in mice after intracerebroventricular treatment by a 26RFa homolog peptide and increased anxiety-like behavior in QRFP-deficient mice [14,15]. 26RFa is present in the blood, showing its increased plasma levels in patients with obesity and diabetes but also in anorexia nervosa [16,17]. No data are available examining possible links of 26RFa to the mood, emotion and aggressive behavior in humans.

Thus, to fill this gap, in the present exploratory study, we analyzed the relevance of 26RFa to behavioral modalities of anxiety, depression and aggression in otherwise healthy humans. For this purpose, the plasma levels of 26RFa were assayed in men with a history of violent aggression, currently serving a sentence in prison (inmates) and compared with those of age-matched men from the general population (controls), all from Norway. The levels of aggressive behavior, anxiety and depression were evaluated using standard questionnaires in both the inmate and control group and analyzed for possible correlations with plasma levels of 26RFa.

## 2. Subjects, Materials and Methods

### 2.1. Study Subjects

This study was approved by the National Research Ethics Committee, case number 2010/792.

### 2.2. Inmates

Thirteen inmates from a high-security prison outside Oslo, Norway, were included in the study. The median age of the inmates was 45 years old, ranging from 27 to 69 years and the mean body mass index (BMI) was 29.7 ± 3.2 kg/m^2^. The inmates were convicts currently serving long-term sentences in preventive detention. The inmates had committed at least one murder or had attempted to commit murder and/or severe sex-related violence, e.g., rape, molesting or grievous bodily harm. One of the included inmates had been released on probation but was re-arrested for the execution of new violent crimes. In Norway, the imposition of preventive detention indicates that the court considers the defendant at high risk for re-offending and, therefore, an imminent threat to society. According to Norwegian law, after serving a minimum term not exceeding ten years, prisoners in preventive detention may ask the court to reconsider their case. Among the 13 prisoners, no endocrine disorders or serious mental illnesses were found when screening for endocrine and psychiatric disorders.

### 2.3. Healthy Male Controls

Data from 19 healthy males without any history of violent crime or previous psychiatric diagnosis were included. The median age of the controls was 42 years old, ranging from 31 to 58 years, and the mean BMI was 26.2 ± 3.5 kg/m^2^. They were either working full-time as drivers, storage managers, leaders, general employees at the fish market in Oslo or as nurses in a hospital.

### 2.4. Exclusions

Female inmates and controls were excluded. In addition, paedophilic inmates and controls with known psychiatric or endocrine disorders—or those on medication for the treatment of such disorders—were excluded. Control subjects with any prior court sentences were also excluded.

### 2.5. Clinical Examination

Before being included, all subjects gave their written consent and had a brief session with a physician and psychiatrist to confirm their current medical status and to screen for previous excluding conditions in their medical history. The inmates and controls were asked whether they had consulted a health professional for any medical or psychiatric disorders. The inmates were asked about the circumstances leading to their arrest and current sentence. Some refused to talk, some did not remember, whereas others spoke freely. Among the latter group, several referenced independently during the conversation, partly remembering what they had done. They went on to say that when serving their sentences with time to reflect, they realized that they knew what they had been doing but felt unable to stop.

### 2.6. Testing Scales

The inmates were tested in the prison with an anxiety and depression scale and an aggression questionnaire. The controls were tested in the hospital. Both groups were tested prior to blood sampling. When compiling the scales, inmates and controls were left alone in an adjacent room with the door open. The scales were examined before being collected so that no items were left unanswered.

### 2.7. The Bryant and Smith Shortened and Refined Aggression Questionnaire (BS-rAQ)

Bryant and Smith (BS) introduced a 12-question refined (r) measurement model of the original Buss–Perry aggression questionnaire (AQ) [18]. The original Buss–Perry AQ had four scales: physical aggression, verbal aggression, anger and hostility. The scale scores were found to correlate with peer nominations of the various kinds of aggression, suggesting the need to assess individual aggressiveness components. Bryant and Smith later found that the four scales did not show adequate common variance (i.e., about 80%). They consequently omitted items with low or multiple loadings and excluded them with reverse-scored wording. This yielded a 12-item, refined four-factor measurement model, which contains fewer than half as many items as the original and is also psychometrically superior.

### 2.8. Hospital Anxiety and Depression Scale (HADS)

To screen for anxiety and depression, the Hospital Anxiety and Depression Scale (HADS scale) was used [19]. This scale is a reliable self-assessment scale developed to screen for symptoms of depression and anxiety in the setting of a hospital medical outpatient clinic. It contains two seven-item subscales, one for anxiety and one for depression. In the current study, the cutoff point was ≥8.

### 2.9. Blood Sampling

After responding to testing scales, the subjects went to the next room, where approximately 5 mL of blood was sampled in an EDTA tube after the puncture of an elbow vein by an authorized bioengineer. No instructions were given to the study persons regarding food or other restrictions prior to the sampling. All blood samples were taken between 900–1200 h in the morning to avoid interfering with the prison’s regular activities. The blood was then transported on ice to the laboratory where the plasma was separated, and aliquots of 2 mL were frozen for the first week at −20 °C, followed by a period at −80 °C until being shipped on dry ice to Rouen, France, where they were kept at −80 °C until the peptide assay.

### 2.10. 26RFa Radioimmunoassay (RIA)

Quantification of 26RFa in plasma samples was carried out using a specific RIA previously developed and validated in the laboratory, detecting both 26- and 43-amino acid forms of the peptide [20]. In brief, for the RIA procedure, each plasma sample was diluted (1:1) in a solution of water/trifluoroacetic acid (99.9:0.1; *v*/*v*). Diluted plasma samples were pumped at a flow rate of 1.5 mL/min through one Sep-Pak C18 cartridge. Bound material was eluted with acetonitrile/water/trifluoroacetic acid (50:49.9:0.1; *v*/*v*/*v*), and acetonitrile was evaporated under reduced pressure. Finally, the dried extracts were resuspended in 0.1 M PBS and assayed for 26RFa.

### 2.11. Statistics

Data were analyzed, and graphs were plotted using GraphPad Prism 5 (GraphPad Software Inc., San Diego, CA, USA). Normality was evaluated by the Kolmogorov–Smirnov (K-S) test. Group differences were analyzed by the Student’s *t*-test or Mann–Whitney (M.W.) test, depending on the normality. Correlations were analyzed using Pearson’s or Spearman’s tests depending on the normality. *p*-value < 0.05 was considered significant. Data are shown as mean ± SEM.

## 3. Results

### 3.1. Behavioral Characteristics of Study Subjects

Compared to men from the general population, inmates have been characterized by elevated parameters of aggressive behavior as reflected by increased BS-rAQ subscale scores of Hostility, Physical aggression and Anger, as well as by the total BS-rAQ score. The BS-rAQ subscale score of Verbal aggression also tended to be higher in inmates vs. controls (Table 1).

The total HAD scores, i.e., evaluating the levels of anxiety and depression, were increased in inmates. This increase was mainly due to the increased HAD subscale score of Anxiety, while the subscale score of Depression showed a tendency for an increase (Table 1).

### 3.2. Plasma 26RFa Concentrations

26RFa peptide immunoreactivity was readably detected in the plasma of all studied subjects showing normal distribution (K-S test *p* > 0.1) in the control group with mean levels of 124.4 ± 14 pg/mL. In the inmate group, the sample distribution did not pass the normality test (K-S *p* < 0.0001) due to the presence of 3 high outliers exceeding by about 20 times the standard deviation of otherwise mean levels of 130.7 ± 41 pg/mL (Figure 1a). A comparison of 26RFa levels between the two groups did not show significant differences (M-W test, *p* = 0.23).

### 3.3. Plasma 26RFa in Relation to Anxiety and Depression

Correlation analysis revealed that plasma levels of 26RFa were positively associated with the HAD Total and its Anxiety subscale scores in the combined group of inmates and controls (Figure 1b). Analyzing the control and inmate groups separately also showed the presence of positive correlations between 26RFa and HAD total and anxiety scores but only in the control group (Table 2). However, after removing the three outliers in the inmate group, the HAD total and HAD Anxiety scores also showed significant positive correlations with plasma 26RFa (Parson’s r = 0.6, *p* = 0.04 and r = 0.7, *p* = 0.02, respectively). No significant correlations between plasma 26RFa and HAD Depression scores were found either in the combined or separate studied groups, with or without outliers (Table 2). Moreover, by stratifying the study subjects into two groups based on their HADS, with a recommended threshold of ≥8, the group with higher scores, i.e., with increased anxiety and depression, showed a significantly higher level of 26RFa (Student’s *t*-test, two-tails *p* = 0.029). This analysis excluded three high outliers, which were all located in the middle of the HAD scores (6, 7 and 9).

### 3.4. Plasma 26RFa and Aggressive Behavior

No significant correlations between plasma levels of 26RFa and BS-rAQ total and subscale scores of aggressive behavior were found either in the combined or separate groups of inmates or controls (Table 3). Nevertheless, after removing the outliers in the inmate group, subscale scores of Hostility correlated significantly with plasma 26RFa within this group (Pearson’s r = 0.72, *p* = 0.009).

## 4. Discussion

This exploratory study is the first attempt to analyze a possible link between the neuropeptide 26RFa and some emotional and behavioral parameters in humans relevant to their anxiety, depression and aggressiveness. For this purpose, plasma levels of 26RFa were assayed in two groups of healthy men: either serving prison sentences or from the general population. The relatively small number of studied subjects, due to the limitation of aggressive inmates available for the research protocol, should be taken into account for a cautious data interpretation, as discussed below.

Importantly, the inmate group was characterized by significantly higher scores of both aggressive behavior and anxiety and showed a tendency for increased depression. From a previous study including these inmates, we also noticed that they were significantly more impulsive than the healthy controls [21]. A particular feature was a sense of negative urgency with a tendency to act rashly and be careless without thought for what might happen. While no significant differences in plasma 26RFa levels were found between the two groups, another main finding of this study was a significant correlation between 26RFa and anxiety.

Neuropeptides have long been known for their modulatory effects on motivated behavior, mood and emotions, making them an attractive target for drug discovery [22]. Their role in anxiety and depression could be related to the modulation of neurotransmitter release, such as inhibition of noradrenaline secretion by galanin [23]. The hypothalamic-pituitary-adrenal (HPA) axis is another putative target for neuropeptides relevant to the regulation of stress response in the context of altered mood, emotion and aggressive behavior [24,25]. Among the neuropeptides reviewed recently for their implication in anxiety and depression, we do not find 26RFa [26,27]. This lack, however, does not mean that 26RFa has no role in emotional regulation but is explained by the fact that only a few studies with relevant data are available. Thus, our study may renew the interest to explore further the involvement of 26RFa in the neurobiological mechanisms of anxiety. For instance, 26RFa produced by the LHA neurons may theoretically modulate the serotonin and noradrenaline signaling, presumably altered in the anxiety and depression [28]. In fact, the mRNA of the 26RFa receptor GPR103 is highly expressed in both the dorsal raphe and locus coeruleus nuclei, i.e., the main serotonin- and noradrenaline-producing sites in the brain, respectively [11]. Also of relevance, microinjection of 26RFa in the locus coeruleus induced an analgesic effect in rats. In contrast, the anxiolytic effect of peptide P550, which is the mouse homolog of neuropeptide 26RFa, involved adrenergic receptors [14,29]. Moreover, central QRFP administration in mice was shown to evoke behavioral arousal, mainly by increased locomotor activity and elevated blood pressure [7]. Although no significant effects of QRFP on anxiety-like behavior in the elevated plus maze test were observed, the same study showed that QRFP increased grooming, which is considered a stress-related response [7].

The positive correlations between plasma levels of 26RFa and anxiety found in the present study do not automatically signify an anxiogenic effect of this peptide. Instead, a compensatory increase of 26RFa to modulate altered monoaminergic and other neurotransmitter releases, which might be the primary underlying mechanism of increased anxiety, can be suggested. In fact, the previous animal studies using the elevated plus maze test showed an anxiolytic effect of 26RFa, while QRFP knock-out mice showed increased anxiety [14,15]. The fact that including the 26RFa outliers from the inmate group in the statistical analysis abolished the significant positive correlation in this group may signify anxiety-unrelated effects of such high 26RFa concentrations.

Although increased anxiety is one of several typical components of major depression, the absence of correlations between 26RFa and depression in the present study points to a rather selective link between 26RFa and anxiety. Moreover, a previous study mentioned a mild effect of QRFP deletion in mice on depressive-like behavior in the tail-suspension test [15].

One of the main aims of this study was to analyze the possible relevance of 26RFa to aggressive behavior. The neurobiology of aggression in humans involves a complex neuronal network modulated by hormonal factors [30]. In particular, the septal area and the VMN have been shown to play key roles in aggressive behavior in rodents. [10]. Several neuropeptides, such as oxytocin and vasopressin, are well-established modulators of aggressive behavior [31]. Therefore, a high level of GPR103 mRNA expression in the rat VMN could be relevant to the putative role of 26RFa in aggressive behavior [11]. Nevertheless, our study did not find significant differences in plasma levels of 26RFa between groups of aggressive inmates and non-aggressive controls. Moreover, no significant correlations between 26RFa levels and scores of aggressive behavior were found in any of these groups. Only after removing the outliers did a subscale of ‘Hostility’ correlate positively with plasma 26RFa in the inmate group. Social anxiety in humans was previously shown to correlate positively with hostility but not with aggression [32]. Thus, the present finding points to a link between 26RFa, anxiety and hostility, probably in the context of social anxiety. Such a connection was not previously explored in rodents which were studied using behavioral tests based on their natural fear of heights and open spaces. Of possible relevance, an earlier study showed that both increased hostility and anxiety in cancer patients correlated positively with plasma 17-hydroxycorticosteroids [33]. This is of interest because GPR103 is abundantly expressed in the adrenal gland cortex and 26RFa was shown to stimulate steroidogenesis both in vivo and in vitro [2,34]. It is, hence, possible that peripheral 26RFa may influence both anxiety and hostility via the modulation of HPA axis activity in the adrenal gland.

We should also admit several limitations of this study. Firstly, a relatively small number of studied subjects makes it underpowered for a typical case-control study. Second, the presence of several confounding factors due to different environmental and nutritional conditions of prison inmates and the control group may influence the 26RFa production. Finally, the presence of a few outliers in the inmate group is puzzling. Taken together, such limitations should classify our study as exploratory. Nevertheless, the data linking 26RFa with anxiety provide the rationale to further explore the functional relevance of this neuropeptide to anxiety and depression, including their clinical forms. The functional significance of GPR103 mRNA expression in the VMN should also be clarified. A possible role in energy metabolism regulation could be expected, although plasma levels of 26RFa did not correlate with BMI in a large cohort of healthy, obese and diabetic subjects [17].

## 5. Conclusions

The present exploratory study compared plasma concentrations of the neuropeptide 26RFa in men from the general population with inmates characterized by increased aggressiveness and anxiety and showed no significant differences between the two groups. However, in both groups, plasma levels of 26RFa correlated positively with anxiety scores but not with those of depression and aggressive behavior, pointing to a rather selective link between 26RFa and anxiety. Future studies will, no doubt, shed more light on the involvement of 26RFa in the regulation of emotional processes in humans.

## Figures and Tables

**Figure 1 brainsci-13-00237-f001:**
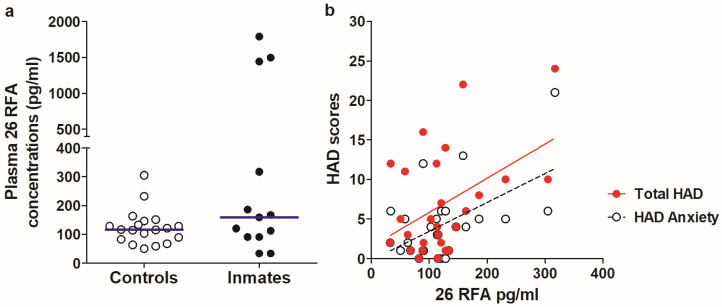
(**a**) Comparison of 26RFa plasma concentration between inmates who had committed violent aggression and men from the general population (Controls). Lines indicate the median levels in both groups. (**b**) Significant positive correlations between plasma concentration of 26RFa and HAD total and HAD Anxiety scores. High outliers in the inmate group have been removed from this correlation plot.

**Table 1 brainsci-13-00237-t001:** Characteristics of aggressive behavior, mood and emotion in control and inmate groups as evaluated by the BS-rAQ and HAD scales. Values corresponding to each scale and subscale scores are shown as mean ± SEM. Group differences were compared using a two-tailed Students’ *t*-test or Mann–Whitney (M-W) test. Significant *p*-values are shown in bold.

Behavioral Scores	Controls (*n* = 19)	Inmates (*n* = 13)	Controls vs. Inmates (*p*-Values)
BS-rAQ Hostility	1.3 ± 0.3	4.2 ± 0.7	***p* = 0.001** (M-W)
BS-rAQ Verbal aggression	2.1 ± 0.3	3.1 ± 0.5	*p* = 0.08
BS-rAQ Physical aggression	1.0 ± 0.3	4.4 ± 0.8	***p* = 0.001** (M-W)
BS-rAQ Anger	1.8 ± 0.4	4.3 ± 0.8	***p* = 0.008** (M-W)
BS-rAQ Total	6.3 ± 0.8	16 ± 2.1	***p* < 0.0001**
HAD Anxiety	2.9 ± 0.5	7.0 ± 1.7	***p* = 0.01**
HAD Depression	1.8 ± 0.6	3.4 ± 0.9	*p* = 0.08 (M-W)
HAD Total	4.8 ± 1.0	10.2 ± 2.1	***p* = 0.02**

**Table 2 brainsci-13-00237-t002:** Correlations between HAD total and subscale anxiety and depression scores with plasma concentrations of 26RFa in the combined group of men which includes inmates who committed aggressive acts, as well as men from the general population as controls and in the latter two groups separately. In the combined group, 3 high levels 26Rfa outliers of the inmate group have been removed. Significant correlation coefficients (Pearson’s or Spearman’s (Sp.) -r), with corresponding *p*-values, are shown in bold.

26RFa Correlations	HAD Subscales		HAD Total
	Anxiety	Depression	
Combined group (*n* = 29)	**r = 0.35** Sp. ***p* = 0.04**	r = 0.14 Sp. *p* = 0.25	**r = 0.46** ***p* = 0.007**
Inmates (*n* = 13)	r = 0.01 *p* = 0.48	r = 0.37 *p* = 0.13	r = 0.16 *p* = 0.3
Controls (*n* = 19)	**r = 0.5** ***p* = 0.02**	r = 0.06 Sp. *p* = 0.4	**r = 0.41** ***p* = 0.04**

**Table 3 brainsci-13-00237-t003:** Correlations between BS-rAQ total and subscale scores with plasma concentrations of 26RFa in the combined group of men which includes inmates who committed aggressive acts, as well as men from the general population as controls and in the two groups separately. In the combined group, 3 high levels 26RFa outliers of the inmate group have been removed. Correlation coefficients (Pearson’s or Spearman’s (Sp.) -r), with corresponding *p*-values, are shown.

26RFA Correlations		BS-rAQ Subscales			BS-rAQ Total
	Hostility	Verbal Aggression	Physical Aggression	Anger	
Combined group (*n* = 29)	r = 0.12 Sp. *p =* 0.26	r = 0.22 *p =* 0.14	r = 0.06 Sp. *p =* 0.38	r = −0.04 Sp. *p =* 0.43	r = 0.05 Sp. *p =* 0.4
Inmates (*n* = 13)	r = −0.14 *p =* 0.32	r = 0.09 *p =* 0.38	r = −0.08 *p =* 0.4	r = −0.04 *p =* 0.46	r = −0.11 *p =* 0.36
Controls (*n* = 19)	r = −0.13 Sp. *p =* 0.3	r = 0.28 *p =* 0.13	r = 0.05 Sp. *p =* 0.4	r = −0.3 *p =* 0.11	r = 0.09 *p =* 0.36

## Data Availability

All data generated or analyzed during this study are included in this article. Further inquiries can be directed to the corresponding author.

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
