# Peer review of "Blood Levels of Neuropeptide 26RFa in Relation to Anxiety and Aggressive Behavior in Humans—An Exploratory Study"

_brainsci, 2023, doi:10.3390/brainsci13020237_

Round 1

Author Response

Reviewer 1

This article provided the interesting findings of possible link between neuropeptide 26RFa and some emotional behavioral parameters in humans relevant to their anxiety, depression and aggressiveness. This is a very well written manuscript. Introduction, method, results, and discussion are well described with enough and wide information. Overall, the critical evaluation of the research within sections is very strong. The section of “Discussion” is excellent and described in detail. It includes many greatly well interpreted information. The Conclusions states the problem and the purpose of review clearly. Effects of neuropeptide 26RFa are very interpreted for better understanding of importance of this article, highlighting its behavioral effects in rodents. The authors present a persuasive argument for a link between the neuropeptide 26RFa and anxiety in humans. Although I am not an expert in such studies, in my opinion the same "conditions" should be provided for blood sampling such as regarding food and other restrictions (these factors can affect the processes) and regarding age, the study could be in a narrower range: “The median age of the inmates was 45 years old, ranging from 27 years to 69 years”. In my opinion, it is very well written manuscript and I support its publication in present form.

Response. We thank the Reviewer for nice comments regarding our study, it is clear that it has limitations which now have been included at the end of the discussion. Indeed, several confounding environmental factors are inherited from this type of studies involving prison inmates, including food, physical activity and social interactions, which all may affect the biological parameters. In fact, we had no way of influencing the inmates’ strict schedule since this was beyond the accepted protocol. We know that the only meal, if consumed, prior to the sampling was an ordinary breakfast. From the other hand, such studies provide a unique opportunity to study possible links between biological parameters and aggressive behavior in humans. In our view, the absence of significant differences of blood levels of 26RFa between aggressive and non-aggressive men clearly show that this regulatory peptide is not involved in the regulation of aggressive behavior. The finding of a significant correlations between 26RFa and anxiety is also of interest because it was present in both groups independently, i.e. should not be affected by the confounding factors due to incarceration.

Reviewer 2 Report

Congrats.

It could be better.

sincerely yours

Author Response

Reviewer 2

Congrats. It could be better. sincerely yours

Response. We thank the Reviewer for the brief and concise evaluation. We did our best to present these data.

Reviewer 3 Report

This is a case control study examining the level of 26RFa, a peptide expressed in the ventromedial hypothalamus (VMH) possibly with orexigenic effects in relation to levels of aggression, anxiety, and depression in 13 inmates and 19 controls. The authors, not surprisingly, did not find differences in the levels of the peptide between the two groups although they found a significant correlation between peptide serum levels and anxiety in the combined sample of cases and controls, although this association was basically driven by controls and not cases.

This is basically a negative study, which is not a weakness itself. The main issue is that this is likely a false negative because the sample is not adequate to detect an association. The authors need to expand their sample collection and test it in sample whose size is established based on a power analysis.

Finally, I am not persuaded that the rationale for studying this peptide in aggression and mood/anxiety is solid. The authors should better motivate why they are investigating this peptide in this phenotype.

Author Response

Reviewer 3

This is a case control study examining the level of 26RFa, a peptide expressed in the ventromedial hypothalamus (VMH) possibly with orexigenic effects in relation to levels of aggression, anxiety, and depression in 13 inmates and 19 controls. The authors, not surprisingly, did not find differences in the levels of the peptide between the two groups although they found a significant correlation between peptide serum levels and anxiety in the combined sample of cases and controls, although this association was basically driven by controls and not cases.

This is basically a negative study, which is not a weakness itself. The main issue is that this is likely a false negative because the sample is not adequate to detect an association. The authors need to expand their sample collection and test it in sample whose size is established based on a power analysis.

Finally, I am not persuaded that the rationale for studying this peptide in aggression and mood/anxiety is solid. The authors should better motivate why they are investigating this peptide in this phenotype.

Response. We thank the Reviewer for the critical comments. We agree that the study would be more convincing if having n=30-40 based on a typical power analysis to obtain moderate effect size. We have admitted it now in the new paragraph about study limitations that is included at the end of discussion. In order to clearly demonstrate this limitation, we also designed the study as “exploratory”. Nevertheless, since it is a first study comparing groups of men with highly significant differences (p<0.0001) of their aggressiveness, such strong stratification even in a relatively small group should reveal potential relevance of a researched biomarkers to the extreme violent phenotype of inmates. Therefore, we believe that it is scientifically justified to present the 26RFa data which did not show statistically significant differences between the groups, and hence, did not support a role of this regulatory peptide in violent aggression.

In fact, in this study we especially searched for the most extreme aggressors since such a selection would give us a status being different from “ordinary” inmates without high hostility and anger issues. Thus, the selection worked in our favor by giving us a relatively defined and homogenous aggressive group to start with. Unfortunately, the recruitment of such study participants was limited by the number of aggressive inmates, including those unwilling to participate within the limited time-space allowed by the research protocol. This, in turn, called for a very careful approach but also an underpowered study.

The rationale to study 26RFa in this context was presented in the introduction. In our view, high level of expression of GPR103 mRNA, the receptor of 26RFa, in the VMN, well-known as the brain “attack center” is a sufficient reason to explore potential link between 26RFa and violent aggression.

Reviewer 4 Report

I would like to thank the authors for the article presented.

-I read the manuscript entitled "Blood levels of neuropeptide 26RFa in relation to anxiety and aggressive behavior in humans".  It is really a very original and interesting topic. It is common in papers for authors to report the positive findings of the study, however for this topic I would ask the authors to report any negative findings as well. For example, did age show a correlation with 26RFa? When the population was split, based on HADS, did the groups show a difference in 26RFa?

-Please add the power analysis of the sample.

- I think it is particularly important for the description of the population to add the BMI (Body Mass Index), if this is available

-I would ask the authors in the correlation table (table 3) to add age and if BMI is available.

-At the end of the discussion add a paragraph with the limitations of the manuscript and thoughts for future studies.

Author Response

Reviewer 4

I would like to thank the authors for the article presented.

Response. We are grateful to the Reviewer to the comments which helped to improve our manuscript and provide point-by-point responses below.

-I read the manuscript entitled "Blood levels of neuropeptide 26RFa in relation to anxiety and aggressive behavior in humans".  It is really a very original and interesting topic. It is common in papers for authors to report the positive findings of the study, however for this topic I would ask the authors to report any negative findings as well. For example, did age show a correlation with 26RFa? When the population was split, based on HADS, did the groups show a difference in 26RFa?

Response. Indeed, as suggested, splitting the study subjects into 2 groups based on their HADS with a recommended threshold of ≥8, the group with higher scores, i.e. with increased anxiety and depression, showed significant higher level of 26RFa (Student’s t-test, 2-tails p=0.029). This analysis excluded 3 high outliers which were all located in the middle of the scale (scores 6, 7 and 9) and are apparently unrelated to emotional regulation. We have added these data in the result section.

-Please add the power analysis of the sample.

Response. The study was exploratory by its nature; it was not performed based on power analysis (see also the response to Reviewer #1).

- I think it is particularly important for the description of the population to add the BMI (Body Mass Index), if this is available.

Response: The mean BMI values have been included in the Methods section.

-I would ask the authors in the correlation table (table 3) to add age and if BMI is available.

Response: Unfortunately, these data were not available. Previous study from our group analyzing a large cohort of healthy, obese and diabetic subjects (n=161) did not find significant correlations between blood 26RFa levels and age or BMI (Prevost et al Diabetes 2015, PMID:25858563). We added this reference to the discussion.

-At the end of the discussion add a paragraph with the limitations of the manuscript and thoughts for future studies.

Response: Thank you for this suggestion, a paragraph summarizing the limitations of the study and future opportunities was included.

Round 2

Reviewer 3 Report

The exploratory angle of the study should reflect on the title and should also determine ad different discussion of the results moderating the statements about the validity of the findings.

Author Response

We thank the Reviewers for this final comment. In the revised version, the exploratory nature of the study was emphasized by including this mention in the title as well as by explanations in the abstract and discussion.